# Combining seasonal malaria chemoprevention with novel therapeutics for malaria prevention: a mathematical modelling study

Lydia Braunack-Mayer[1,2], Josephine Malinga[1,3], Narimane Nekkab[4], Sherrie L. Kelly[4], Jörg J. Möhrle[4,5], Melissa A. Penny[1,3]*

1 The Kids Research Institute Australia, Nedlands, Australia, 2 Institute of Social and Preventive Medicine, University of Bern, Bern, Switzerland, 3 Centre for Child Health Research, The University of Western Australia, Crawley, Australia, 4 Swiss Tropical and Public Health Institute, Allschwil, Switzerland, 5 University of Basel, Basel, Switzerland

* melissa.penny@uwa.edu.au

## Abstract

Vaccines, monoclonal antibodies, and long-acting injectables are being developed to prevent *Plasmodium falciparum* malaria. These therapeutics may target multiple stages of the parasite life cycle; evidence is needed to articulate their benefits with chemoprevention and prioritise candidates for clinical development. We used an individual-based malaria transmission model to estimate the health impact of combining new therapeutics with seasonal malaria chemoprevention (SMC). Our modelling framework used emulator-based methods with models of pre-liver and blood stage therapeutic dynamics. We evaluated the benefit of combining therapeutics with SMC in children under five by estimating reductions in the cumulative incidence of uncomplicated and severe malaria, relative to SMC or the new therapeutic alone, during and five years after deployment. Results showed that new therapeutics may require extended pre-liver stage duration or multi-stage activity to combine with SMC. For three SMC cycles in a high transmission setting, a pre-liver stage therapeutic with partial initial efficacy (>50%) required a protection half-life >230 days to reduce cumulative severe cases by >23% during interventions, and >5% five years after deployment stopped. Longer protection was needed when combined with four or five SMC cycles. And, combining SMC with a multi-stage therapeutic increased public health impact both during and after deployment. These results indicate that combining SMC with malaria therapeutics active against multiple stages of the parasite life cycle can improve the effectiveness of SMC, and highlight the need for clinical development to prioritise multi-stage therapeutics for improved malaria prevention in children.

**Data availability statement:** All code and datasets are available on a GitHub repository at https://github.com/lydiab-mayer/modelling-malaria-combination-interventions/releases/tag/v1.0.0.

**Funding:** LBM, JM, NN, SLK were funded by the Gates Foundation (INV-002562 to MAP, https://www.gatesfoundation.org/). MAP acknowledges support from the Swiss National Science Foundation (SNF Professorship PP00P3_170702 and PP00P3_203450, https://www.snf.ch/). JJM was an employee of Medicines for Malaria Venture (https://www.mmv.org/). Together with other experts, representatives from the Gates Foundation contributed to discussions regarding use cases for novel prevention therapeutics. The funders had no role in the study design, model simulation, data analysis, result interpretation, decision to publish, or manuscript preparation.

**Competing interests:** I have read the journal's policy and the authors of this manuscript have the following competing interests: MAP was part of the WHO Guidelines Development Group for Malaria Chemoprevention (2020–21). JJM was an employee of Medicines for Malaria Venture. All other authors declare no competing interests.

## Author summary

Preventive interventions, such as seasonal malaria chemoprevention (SMC), have reduced the burden of malaria in children living in regions with seasonal malaria transmission. Our study asks how best to combine new malaria prevention products, such as vaccines, monoclonal antibodies, and long-acting injectable drugs, with existing tools like SMC. Understanding which product characteristics make these combinations most useful is important to be able to guide their development and deployment. In this study, we used mathematical modelling to explore how different types of malaria therapeutics might be combined with SMC to improve health outcomes in children. We found that therapeutics targeting early stages of infection need to provide long-lasting protection to offer added value alongside SMC. However, combining SMC with new tools almost always led to better outcomes than using the new tools alone, highlighting the importance of continuing to provide SMC. We also found that therapeutics that act against multiple stages of the malaria parasite could offer greater protection against severe disease. These findings can help inform priorities for product developers, funders, and policymakers working to reduce the burden of malaria through smarter use of new and existing tools.

## Introduction

The first five years of a child's life play a pivotal role in shaping their long-term health and development, and early childhood illness can significantly impact a child's ability to learn and succeed later in life [1]. Nevertheless, in regions of the world where *Plasmodium falciparum* malaria is endemic, this crucial period in a child's life coincides with their highest vulnerability to malaria infection. The World Health Organization estimates that, in 2024, approximately 434 000 children under the age of five years died of malaria in Africa [2]. In Sub-Saharan regions of Africa, public health interventions, such as insecticide treated nets [3] and chemoprevention [4], are crucial to provide children with tools that prevent malaria during these formative years.

In a number of African regions, the high-risk period for *P. falciparum* malaria transmission coincides with the rainy season. The World Health Organization recommends seasonal malaria chemoprevention (SMC), which clears existing infections and prevents new cases through monthly cycles of an antimalarial drug [4]. The drug combination sulfadoxine-pyrimethamine with amodiaquine (SP-AQ) has been widely used for SMC in children [4], achieving a 88.2% (95% CI 78.7–93.4) mean reduction in the incidence of clinical malaria in children younger than five years when deployed at scale [5]. However, children do not always comply with the full treatment course of SP-AQ, high levels of coverage are not always achieved, and SMC's effectiveness may be compromised due to the spread of partial *P. falciparum* resistance to SP [5,6]. These gaps leave some children at risk of malaria infection during a crucial period in their development.

In 2021, the first randomised control trial for SMC with SP-AQ in combination with malaria vaccination was completed [7]. This study combined up to five cycles of an established chemoprevention intervention with seasonal deployment of the first approved malaria vaccine, RTS,S/AS01E (RTS,S). SP-AQ acts to clear blood stage parasites and prevent liver stages from emerging into the blood, while RTS,S is a pre-liver stage vaccine that prevents liver stage infection by targeting surface antigens of sporozoite parasites. The combination of a pre-liver stage vaccine with SMC was found to be superior to SMC alone [7–9]. SMC has also been studied in combination with the R21/Matrix-M malaria pre-liver stage vaccine [10]. And, modelling studies have identified the potential for substantial public health benefits from combining RTS,S with SMC [11]. These successes demonstrate that a pre-liver stage vaccine can be used to strengthen the effectiveness of malaria chemoprevention and to achieve significant reductions in childhood burden. However, it is not yet known whether alternative therapeutics will provide similar benefits, or whether therapeutics that target additional stages of the parasite life cycle could be more effective.

Recent developments in vaccines, monoclonal antibodies, and long-acting injectable drugs suggest opportunities to combine chemoprevention with therapeutics other than pre-liver stage vaccines, including with therapeutics active against multiple stages of the parasite life cycle. The first phase two clinical trial results for a pre-liver stage monoclonal antibody are now available [12]. This class of therapeutic can provide high levels of protection following a single dose, making it a likely candidate for seasonal deployment. But concerns remain regarding the cost-effectiveness of monoclonal antibodies when deployed at scale [13,14]. Injectable formulations of existing antimalarial drugs, such as atovaquone-proguanil, may offer children three to four months of protection at a lower cost than monoclonal antibodies [14–17]. These re-formulations can target multiple stages of the parasite life cycle by, for example, combining drug partners with both liver stage and blood stage activity. There are also multi-stage vaccine candidates in development, including whole-sporozoite [18], blood stage [19], and transmission blocking vaccines [20]. Each of these novel therapeutics could be combined with SMC to improve the effectiveness of childhood malaria interventions. But each of these novel therapeutics has different profiles, encompassing differences in the levels of protection provided immediately following administration, the length of protection afforded, and their rate of efficacy decay. It is not yet known how these differences in therapeutic properties will relate to their public health impact once combined with SMC.

In this modelling study, we explored and defined the essential properties needed for a seasonally deployed, novel malaria therapeutic to increase the effectiveness of SMC throughout childhood. By novel therapeutic, we refer to a malaria vaccine, monoclonal antibody, or long-acting injectable drug that has not yet been approved for use in children. Our research combined an established, individual-based malaria transmission model with statistical modelling techniques, quantifying the therapeutic properties required for a new therapeutic to contribute additional reductions in cumulative uncomplicated and severe cases in children, in comparison to the standard of care. Our results encompassed vaccines, monoclonal antibodies, and novel long-acting therapeutics with both activity against a single-stage of the parasite life cycle (either pre-liver or blood stage) and activity against multiple stages (pre-liver combined with blood stage, referred to as multi-stage therapeutics). Through this evidence, we have aimed to provide funders, policymakers, and developers with guidance to inform candidate selection of malaria therapeutics with adequate properties to meet public health goals in advance of phase two and three clinical trials.

## Materials and methods

### Malaria transmission model

We used an established model of malaria transmission to predict the impact of combining SMC with seasonal deployment of a novel therapeutic. This individual-based model, called OpenMalaria (https://github.com/OpenMalaria-Org/openmalaria), has been fully described previously [21]. OpenMalaria combines multiple, smaller models that each represent a component in the malaria transmission lifecycle. Model components simulate the dynamics of mosquito populations, malaria parasitaemia in the course of an infection, onward transmission, and the processes leading to clinical illness, severe disease, and death (Table A in S1 Text). This model has been validated to historical data on a range of

epidemiological outcomes, including patterns of age-specific prevalence and incidence, and age-specific rates for malaria mortality and hospitalisation [21].

Importantly for this study, OpenMalaria includes detailed within-host models that represent parasite dynamics across multiple stages of the *P. falciparum* life cycle. The model simulates the probability of liver stage infection following an infectious mosquito bite, the resulting blood stage parasite density, and the probability of onward transmission. Therapeutics can target each of these stages independently. As a result, synergistic effects of combining interventions on population-level outcomes, such as those observed in the clinical trial of RTS,S/AS01 and SMC [7–9], can arise from the underlying model mechanisms, rather than being imposed at an individual level through assumed interaction terms or fixed efficacy profiles. This allows interactions between therapeutics, including vaccines and drugs, to emerge at the population level as a consequence of their combined effects on parasite development and host immunity.

Finally, OpenMalaria's model components capture the dynamics of acquired immunity to the asexual blood stages of *P. falciparum* infection. Acquired immunity is represented as a function of an individual's cumulative density of asexual parasitaemia since birth, including maternal immunity, and their total number of prior infections. Infection densities are drawn from parasitaemia records of malaria-naive patients infected with *P. falciparum* for treatment of neurosyphilis [22–24]. The number of prior infections influences pre-erythrocytic immunity, and both prior infections and cumulative asexual parasitaemia influence blood stage immunity. Immunity acts by reducing an infected host's parasite densities and affecting the probability that parasitaemia will reach the specific parasite densities, or pyrogenic thresholds, that trigger symptoms of uncomplicated and severe malaria [23,24].

### Intervention dynamics and deployment

To model SMC with SP-AQ, we represented the drug combination with both preventative and curative effects. SP-AQ was assumed to prevent new infections by blocking liver-stage parasite emergence and to cure existing infections by clearing blood and liver-stage parasites at the time of administration. These assumptions followed our review of the mechanisms of action of SP-AQ for malaria prevention [25]. Efficacy decay for the preventative action was calculated with a Weibull function: $E(t) = E(0) \times \exp(-(\frac{t}{L})^k \times \log(2))$, where $E(0)$ was the initial probability of preventing infection at the time of administration, $L$ the scale parameter was the number of days until this probability reached half of its initial value, and $k$ the shape parameter influenced the rate of efficacy decay.

To model novel malaria prevention therapeutics, we simulated a range of malaria vaccine, monoclonal antibody, and long-acting injectable drug candidates by combining modelling their activity against different stages of the *P. falciparum* parasite life cycle. Pre-liver stage therapeutics were assumed to reduce the probability of infection following exposure, while blood stage therapeutics killed circulating parasites. Multi-stage therapeutics were modelled by combining both components. As for SP-AQ, the efficacy of each component was represented with a Weibull function of time following administration. These model parameters are illustrated in Fig A in S1 Text.

For each therapeutic (SP-AQ, pre-liver stage, blood stage, and multi-stage profiles), Table 1 outlines the modelled range of parameters used. For SMC, model parameter values were previously calibrated to data from a randomised non-inferiority trial of SMC with SP-AQ [26,27]. For novel therapeutics, we sampled 500 sets of parameter values using Latin hypercube sampling with a uniform probability distribution. Parameter ranges were chosen to represent the broadest possible range of therapeutic profiles, from a highly efficacious, long-acting vaccine (initial efficacy 100%, 500 day protection half-life, exponential decay shape of one) to a low-efficacy, short-acting drug (30% initial efficacy, 30 day protection half-life, rapid decay shape of nine). By simulating outcomes for this broad parameter space of protective properties, we were able to explore the impact of combining SMC with a full spectrum of possible therapeutics. This space aimed to encompass the properties of future novel vaccines, monoclonal antibodies, and long-acting injectable drugs.

We then used our individual-based model of malaria transmission to simulate the deployment of SMC with SP-AQ, both alone and in combination with novel malaria prevention therapeutics. The novel therapeutic (pre-liver stage, blood stage,

**Table 1. Model assumptions for dynamics of therapeutic efficacy and decay.**

| Therapeutic component | Modelled biological action | Efficacy decay function | Initial efficacy E(0)[a] | Protection half-life L (days)[b] | Decay shape k[c] |
|---|---|---|---|---|---|
| SP-AQ | Probability of preventing liver stage infection from establishing | $E(t) = E(0) \times e^{-\left(\frac{t}{L}\right)^{k} \times \log(2)}$ | 100%[d] | 31.1 [d] | 5.34 [d] |
| Pre-liver stage | Probability of preventing liver stage infection from establishing | $E(t) = E(0) \times e^{-\left(\frac{t}{L}\right)^{k} \times \log(2)}$ | 30 to 100%[e] | 30 to 500 days [e] | 0 to 10 [e] |
| Blood stage | Proportion of circulating blood stage parasites killed | $E(t) = E(0) \times e^{-\left(\frac{t}{L}\right)^{k} \times \log(2)}$ | 30 to 100% [e] | 30 to 500 days [e] | 0 to 10 [e] |
| Multi-stage | Combination of pre-liver and blood stage components | Both functions applied independently to each stage | See above | See above | See above |

[a] E(0) is the efficacy at the time of administration (probability of preventing infection or proportion of parasites cleared).

[b] L is the protection half-life: time in days until efficacy decays to 50% of E(0).

[c] k controls the shape of the decay: k < 1 indicates slow (long-tailed) decay, k > 1 results in rapid, sigmoidal decay.

[d] Parameter values were previously calibrated to data from a randomised non-inferiority trial of SMC with SP-AQ. [26,27]

[e] 500 parameter values were sampled using Latin hypercube sampling with a uniform probability distribution.

SP-AQ = sulfadoxine-pyrimethamine + amodiaquine.

multi-stage) was deployed together with the distribution of SP-AQ in the first cycle of SMC, representing a scenario where a child receives a vaccine, monoclonal antibody or long-acting injectable in combination with receipt of SP-AQ. All interventions were deployed to children aged between three and 59 months of age. Importantly, as described above, we did not make any assumptions about drug-drug or vaccine-vaccine interactions, synergies or antagonisms between the two components of multi-stage therapeutics; any synergistic effects on clinical outcomes at population-level emerged from the underlying within-host intervention dynamics of OpenMalaria.

### Intervention model validation

In a previous study, we validated our model's ability to accurately represent the effectiveness of SMC in combination with a pre-liver stage vaccine [28]. We used OpenMalaria to replicate outcomes from the only available clinical trial evaluating this combination: the individually randomised trial of RTS,S with or without SMC conducted in Mali and Burkina Faso [7,8]. Using data from the first two years of the trial, we calibrated vaccine booster efficacy and the therapeutic properties of SP-AQ to match observed clinical incidence in all three trial arms: SMC alone, RTS,S alone, and the combination. This was done with a Bayesian optimisation framework within OpenMalaria, taking initial efficacy estimates for RTS,S after the three-dose primary series (91.1%) and a half-life of 7.3 months from previous modelling studies and clinical trial data [29]. These values were used as inputs, and boosting efficacies were estimated by optimising model predictions against trial data.

In this previous study, OpenMalaria successfully reproduced monthly clinical incidence and hazard ratios for all arms of the trial in both countries. For RTS,S booster doses, the model estimated that a 12-month post-primary booster restored efficacy to 78.96% (95% CI 70.74 – 87.18%) in the vaccine-only arm and 77.45% (95% CI 68.26 – 86.62%) in the combination arm. A second booster restored maximum efficacy to 57.31% (95% CI 44.05 – 70.56%) and 48.76% (95% CI 34.57 – 62.95%), in the vaccine only and combination arms respectively. These values matched closely with observed outcomes, and our model captured both temporal trends and relative effectiveness across trial arms.

### Outcome measures

Our primary outcome measures captured reductions in the cumulative incidence of uncomplicated and severe malaria throughout childhood. We followed children born in the first year of intervention delivery and evaluated cumulative incidence at five (as the intervention ends) and ten years old (five years after interventions have ended). As illustrated in Fig 1, we

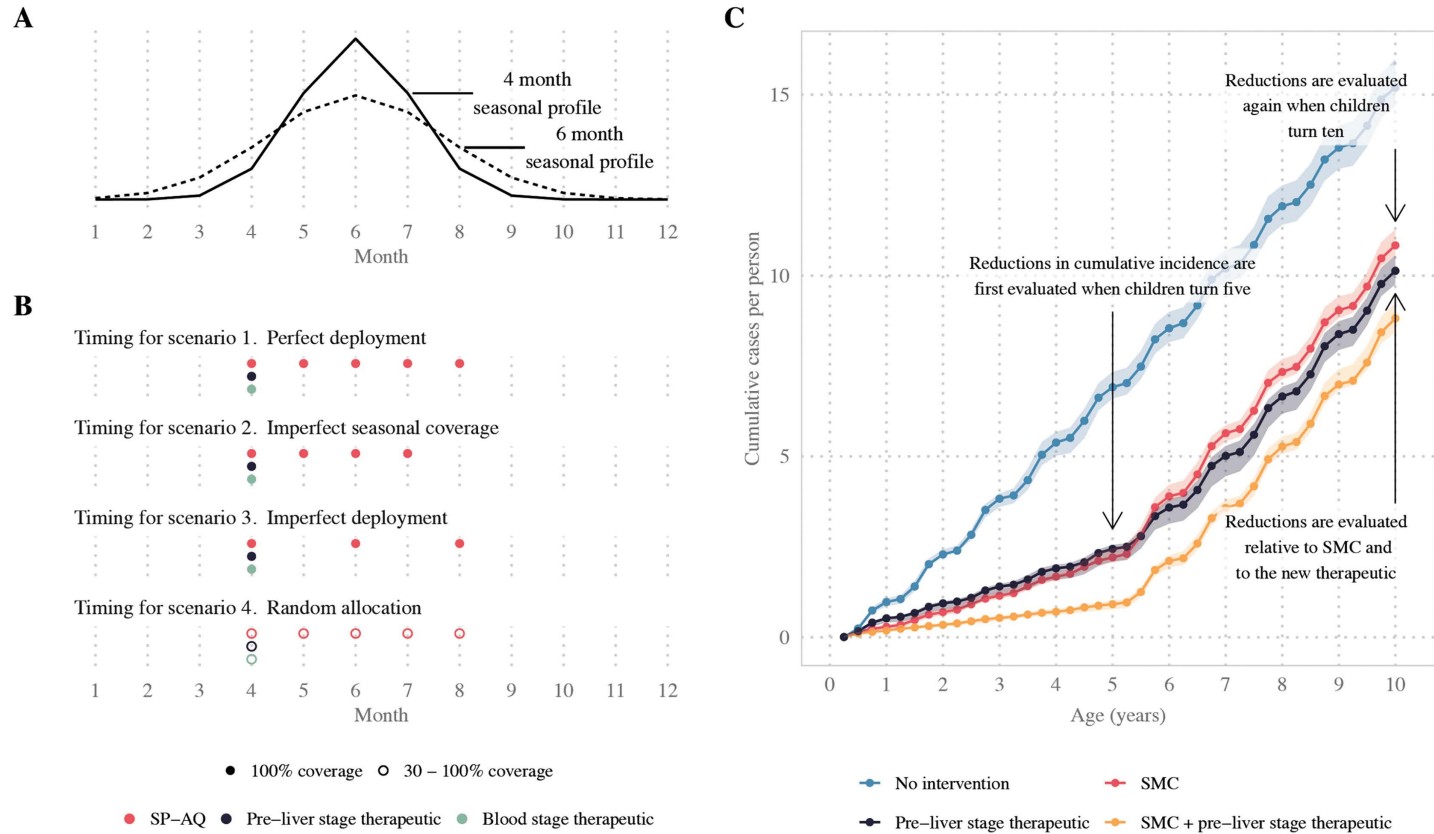

**Fig 1. Illustration of seasonality profiles, intervention timing, and primary outcome measures.** Panel A illustrates the two seasonal profiles modelled, where 70% of malaria cases occur within a four- or six-month period and where the y-axis represents the average annual entomological inoculation rate (EIR). Panel B illustrates interventions deployed according to four seasonal deployment scenarios. Scenario 1. Perfect deployment: All children aged between three and 59 months had perfect compliance to five yearly SMC cycles. Scenario 2. Imperfect seasonal coverage: All children aged three to 59 months had perfect compliance to four yearly SMC cycles but missed the final cycle. Scenario 3. Imperfect deployment: All children aged three to 59 months had perfect compliance to yearly SMC cycles one, three, and five. Scenario 4. Random allocation: 30% to 100% of children were randomly selected to receive each of five yearly SMC cycles. Panel C illustrates the evaluation of our primary outcome measures. We tracked a cohort of children born in the first year of intervention deployment, and evaluated cumulative incidence when children were five and ten years old. We then calculated the reductions in the cumulative incidence of uncomplicated and severe malaria attributed to the combination of SMC and the novel therapeutic. Reductions were evaluated relative to two counterfactual scenarios: regular deployment of SMC alone, and seasonal deployment of a novel therapeutic alone. Shaded regions indicate the minimum and maximum cumulative cases per person observed across ten stochastic replicates of simulations from the individual-based malaria transmission model.

then calculated the reductions in the cumulative incidence of uncomplicated and severe malaria attributed to the combination of SMC with SP-AQ and the novel therapeutic. Reductions were evaluated relative to two counterfactual scenarios: regular deployment of SMC alone, and seasonal deployment of a novel therapeutic alone (Section 1 in S1 Text). We report these reductions as the percentage of burden averted by the combination relative to a standard of care; we do not compare to a no-intervention scenario.

## Simulation and model scenarios

The malaria transmission model was used to simulate cumulative case outcomes for a range of transmission and health system scenarios. Each model scenario captured a unique combination of the following characteristics: a low (10%)

or high (50%) probability of seeking first-line treatment for clinical malaria over 14 days post-infection; 70% of malaria cases occurring within a four- or six-month transmission season (Fig 1); six levels of transmission intensity, ranging from low-moderate to very high baseline annual prevalence. Annual prevalence was measured as the prevalence rate of *P. falciparum* infections detectable by rapid diagnostic test in children aged two to ten years old ($Pf$PR$_{2-10}$) when no intervention was deployed. Since $Pf$PR$_{2-10}$ is calculated from simulated outcomes, it varies with both model scenario and stochasticity.

We also modelled four different deployment scenarios for SMC, as illustrated in Fig 1. In the perfect deployment scenario, all children aged between three and 59 months had perfect compliance to five yearly SMC cycles. This scenario represented an ideal clinical trial where every child is perfectly adherent to the SP-AQ chemoprevention regimen. In the imperfect seasonal coverage scenario, all children aged three to 59 months had perfect compliance to four yearly SMC cycles but missed the final cycle. This scenario represented a setting where the number of SMC cycles may not provide protection for the full transmission season. In the imperfect deployment scenario, all children aged three to 59 months had perfect compliance to yearly SMC cycles one, three, and five. This scenario represented a setting where children systematically miss out on multiple SMC cycles. In the random allocation scenario, 30% to 100% of children were randomly selected to receive each of five yearly SMC cycles. This scenario represented a setting where children randomly miss out on one or more SMC cycles. In all four deployment scenarios, the novel prevention intervention was co-deployed seasonally together with the first cycle of SMC. By varying deployment, we were able to quantify differences in the novel intervention's benefit when children are more or less covered by SMC.

## Statistical analysis

We followed an existing mathematical framework for defining essential therapeutic characteristics for malaria prevention interventions [30]. This framework has previously been used to explore determinants of malaria intervention impact for long-acting injectable drugs [26] and for novel seasonal malaria chemoprevention therapeutics [31].

In brief, following our methodology (Section 1 and Fig B in S1 Text), we first generated a Latin hypercube sample of 500 values for the novel intervention's parameters: therapeutic initial efficacy, protection half-life, and decay shape. For each sample, we used our individual-based malaria transmission model to simulate outcomes with ten stochastic replicates. Heteroskedastic Gaussian process regression models [32] were then fitted to predict primary outcome measures (reductions in cumulative uncomplicated and severe cases at five and ten years old) as a function of intervention inputs (initial efficacy, protection half-life, and decay shape). These models were used to emulate the relationship between therapeutic properties and our primary outcome measures without additional simulation from the individual-based model, which would have required intensive computation. Emulator performance was evaluated against a 10% hold-out set. Finally, for each model scenario, we quantified the impact of a small change in each key therapeutic property on variation in our primary outcome measures by performing a non-parametric sensitivity analysis using the Sobol-Jansen method [33,34]. Sobol total-order indices were computed for two Latin hypercube samples of 50 000 intervention parameters. All analyses were conducted in R [35].

## Results

### Combining SMC with seasonal deployment of a pre-liver stage therapeutic

We first simulated cumulative case outcomes among children for exemplar malaria therapeutic profiles: one pre-liver stage therapeutic with long duration and high efficacy (protection half-life 354 days, 90% initial efficacy) and a second with shorter duration and partial efficacy (protection half-life 120 days, 50% initial efficacy). This allowed us to explore the therapeutic properties needed for a pre-liver stage therapeutic to add benefit to SMC with SP-AQ. Fig 2 depicts the cumulative incidence curves predicted by our model for two hypothetical pre-liver stage therapeutics. First, we saw that both therapeutics contributed to reductions in cumulative uncomplicated and severe cases by the time children were five years old (reductions >10%). This result confirmed the positive effects observed in the randomised control trial of seasonal vaccination with RTS,S in combination with SMC, [8,9] and the R21 phase three clinical trial. [10]

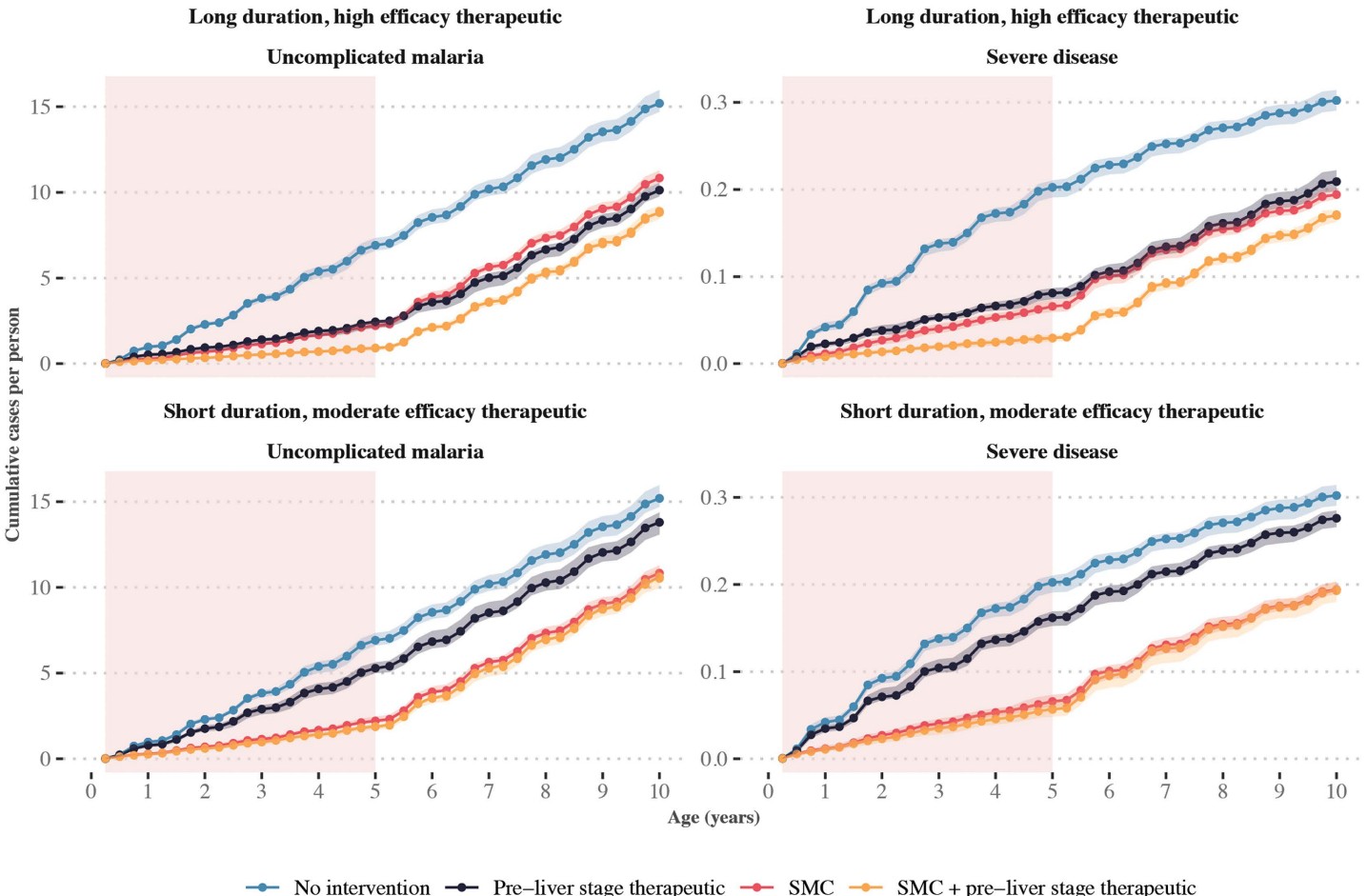

**Fig 2. Predicted cumulative case curves when seasonal deployment of two exemplar pre-liver stage therapeutics are combined with SMC.** The median number of cumulative cases per year of age were calculated when children aged three to 59 months received one of the following interventions: no intervention, three SMC cycles (imperfect deployment scenario), seasonal deployment of a pre-liver stage therapeutic, or the combination of three SMC cycles and the pre-liver stage therapeutic. Top panels include deployment of a pre-liver stage therapeutic with a protection half-life of 354 days, 90% initial efficacy, and decay shape parameter of 1. Bottom panels include a pre-liver stage therapeutic with a protection half-life of 120 days, 50% initial efficacy, and decay shape parameter of 1. The pink shaded regions indicate age-eligibility for SMC and the novel therapeutic. Shaded regions indicate the minimum and maximum cumulative cases per person observed across stochastic replicates of simulations from the individual-based malaria transmission model. Model results are shown for a scenario with transmission intensity corresponding to 32% $PfPR_{2-10}$, where 75% of malaria cases occur within six months of the year and the probability of seeking first-line treatment for clinical malaria over 14 days is low (10%).

However, we saw that pre-liver stage therapeutics needed to have a long duration and high efficacy to maintain their benefits in comparison to SMC as children aged. The exemplar therapeutic with shorter duration and partial efficacy (Fig 2) led to no observable difference in cumulative uncomplicated cases by the time children were eight years old. Furthermore, we observed that the combination of SMC with SP-AQ and the pre-liver stage therapeutic was more beneficial than the pre-liver stage therapeutic alone throughout childhood. Similar patterns were observed for other SMC deployment scenarios (Fig C, D in S1 Text) and across the full range of pre-liver stage therapeutic properties (Fig E in S1 Text). Together, these findings indicate that pre-liver stage therapeutics may need a long duration to limit the potential for delayed malaria when added to SMC and may not fully replace SMC's benefits.

In addition to a long duration, we found that pre-liver stage therapeutics needed a long tail of protection to maintain reductions in cumulative burden as children aged. Higher reductions in cumulative cases by ten years of age were observed for pre-liver stage therapeutics with a long tail of protection, encompassing exponential and biphasic decay profiles (decay shape ≤1, Fig 3), in comparison to sigmoidal decay (decay shape >1). For example, in Fig 3, deploying a pre-liver stage therapeutic with a decay shape parameter between 0.6 and 0.8 was associated with a 9.9% median reduction (interquartile range 5.5-16.0% across the parameter range for protection half-life and initial efficacy) in cumulative uncomplicated cases by the time children were ten years old, evaluated relative to SMC alone. In contrast, deploying a therapeutic with a decay shape between 4.0 and 4.2 led to a lower median reduction of 5.3% (interquartile range 1.8-10.0%). Similar findings were observed across deployment scenarios (Fig F, G, H in S1 Text).

**A. Pre–liver stage parameter relationships with cumulative case outcomes by five years old**

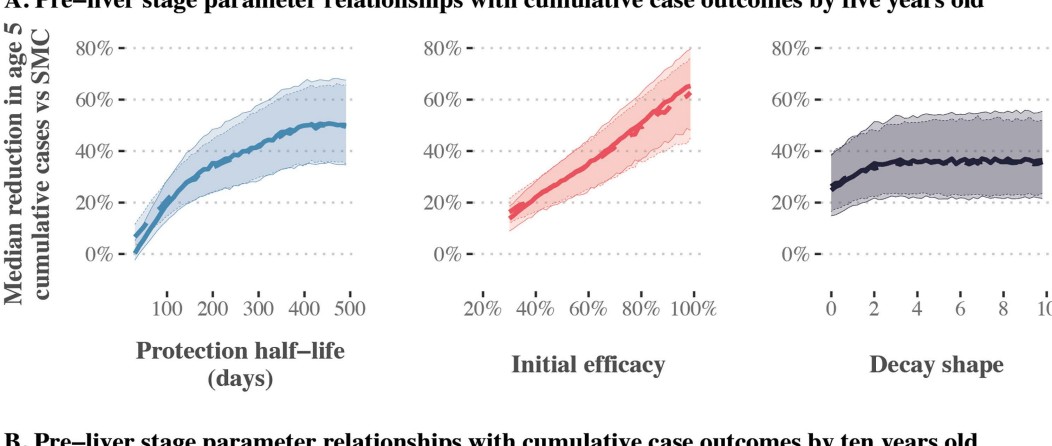

**B. Pre–liver stage parameter relationships with cumulative case outcomes by ten years old**

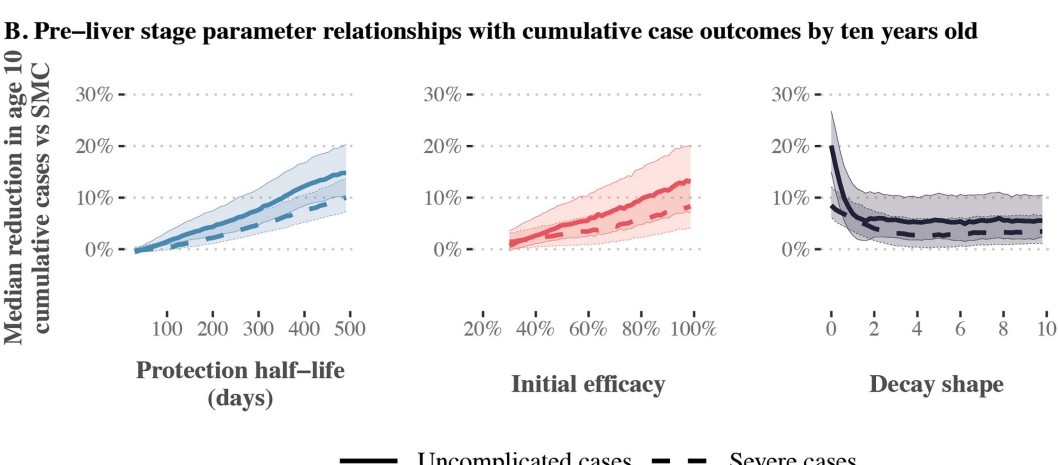

—— Uncomplicated cases  – – Severe cases

**Fig 3. Gaussian Process regression emulator predictions for the relationship between pre-liver stage therapeutic properties and reductions in cumulative uncomplicated and severe cases relative to SMC alone.** Results show the imperfect deployment scenario, where children aged three to 59 months received three SMC cycles. Transmission is high (32% $PfPR_{2-10}$), 75% of malaria cases occur within six months of the year, and the probability of seeking first-line treatment for clinical malaria over 14 days is low (10%). Each panel shows the median reduction in cumulative uncomplicated (solid lines) or severe cases (dashed lines) achieved by combining seasonal deployment of a pre-liver stage therapeutic with SMC, relative to cumulative cases when SMC is deployed alone. Median reductions are calculated by dividing the parameter range for the therapeutic property shown on the x-axis (protection half-life, initial efficacy, decay shape) into 51 segments, and calculating the median outcome for each given segment across all other parameter values. Shaded regions represent the 25th and 75th percentiles of the corresponding reductions. Predictions are shown separately for cumulative case outcomes at five (top panels) and ten years old (bottom panels). SMC = seasonal malaria chemoprevention.

Our results showed that co-deploying SMC with a pre-liver stage therapeutic is likely to have the most impact in settings where SMC does not fully protect children. For example, for a pre-liver stage therapeutic with 50% initial efficacy and a long tail of protection (biphasic decay shape 0.7) deployed as in our imperfect SMC deployment scenario, with three yearly cycles of SMC, a target 5% reduction in cumulative severe cases by ten years old required a protection half-life of at least 237 days (Fig 4). For the same therapeutic profile, a protection half-life of at least 237 days corresponded to a reduction in severe cases by five years old of more than 23%. When the same therapeutic was combined with five SMC cycles, as in our perfect SMC deployment scenario, however, a 5% reduction required a protection half-life of at least 340 days. Results also indicated that higher initial efficacy could be traded for shorter duration requirements; in Fig 4, the target 5% reduction in cumulative severe cases by ten years old could be achieved with a protection half-life of 153 days if pre-liver stage initial efficacy was 70%. Similar dynamics were observed across seasonal transmission settings and levels of access to care, although with higher impact observed when the season was longer (six month vs. four month seasonal profile) and when access to care was higher (50% vs. 10% access to care). (Fig I in S1 Text)

## Combining SMC with seasonal deployment of a blood stage therapeutic

Our results indicate that combining a blood stage therapeutic with SMC is likely to reduce the occurrence of severe malaria but may not reduce uncomplicated cases. When a blood stage therapeutic was combined with SMC, increasing a

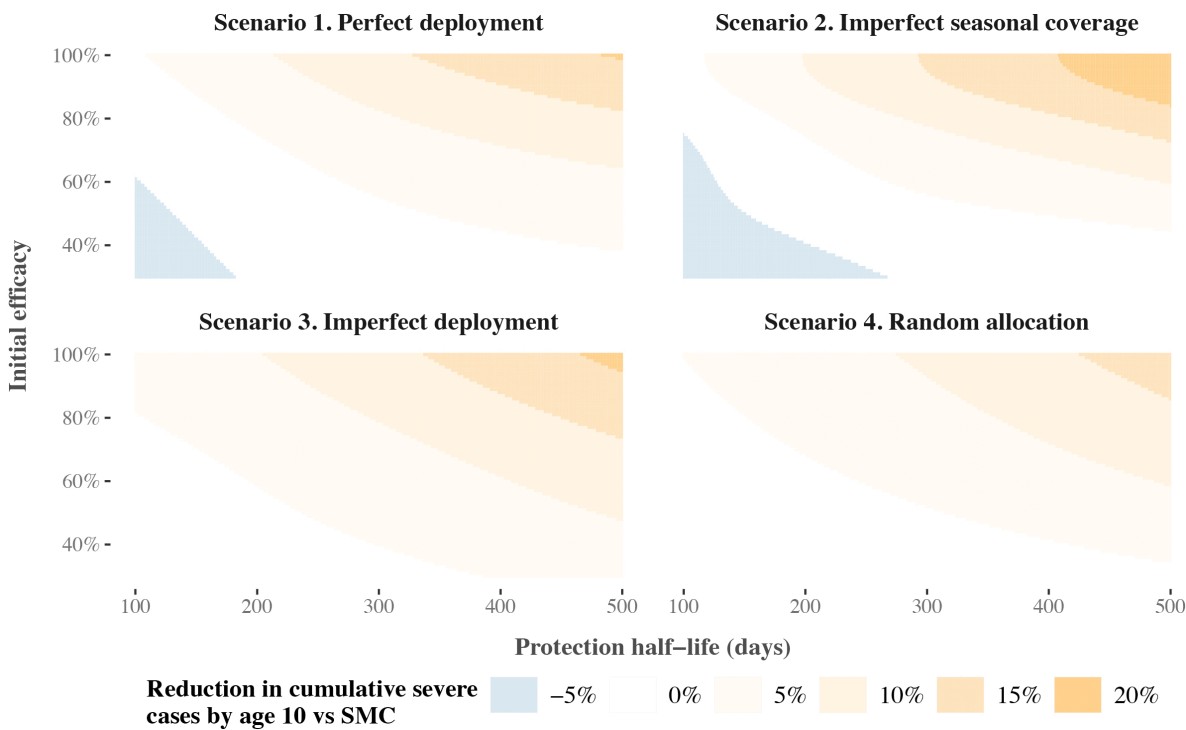

**Fig 4. Predicted relationships between pre-liver stage therapeutic properties and reductions in cumulative severe cases relative to SMC alone, compared across SMC deployment scenarios.** Grid squares indicate the predicted reduction if a pre-liver stage therapeutic with the given protection half-life (x-axis) and initial efficacy (y-axis) were deployed in addition to SMC, assuming a long tail of protection (decay shape parameter 0.7). Reductions are rounded to the nearest 5%. Results show a scenario where transmission is high (32% $PfPR_{2-10}$), 75% of malaria cases occur within six months of the year, and the probability of seeking first-line treatment for clinical malaria over 14 days is low (10%). Panels depict predicted relationships for four different SMC deployment scenarios. In deployment scenario 4, SMC coverage is fixed at 70%. SMC = seasonal malaria chemoprevention.

therapeutic's initial efficacy or protection half-life led to limited change in reductions in uncomplicated malaria. As shown in Fig 5, a near-perfect initial blood stage efficacy of 95.8% was required to achieve a positive median reduction in cumulative uncomplicated cases by ten years old. In contrast, a 10% reduction in cumulative severe cases could be achieved with an initial blood stage efficacy of 86%. Similar trends were observed across SMC deployment scenarios (Figs K, L, M in S1 Text).

**A. Blood stage parameter relationships with cumulative case outcomes by five years old**

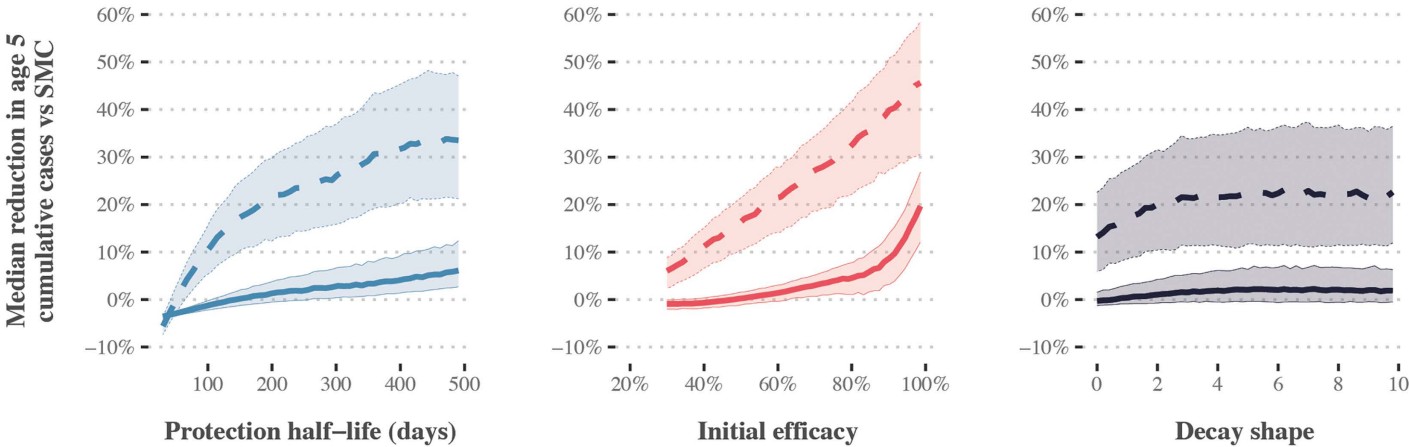

**B. Blood stage parameter relationships with cumulative case outcomes by ten years old**

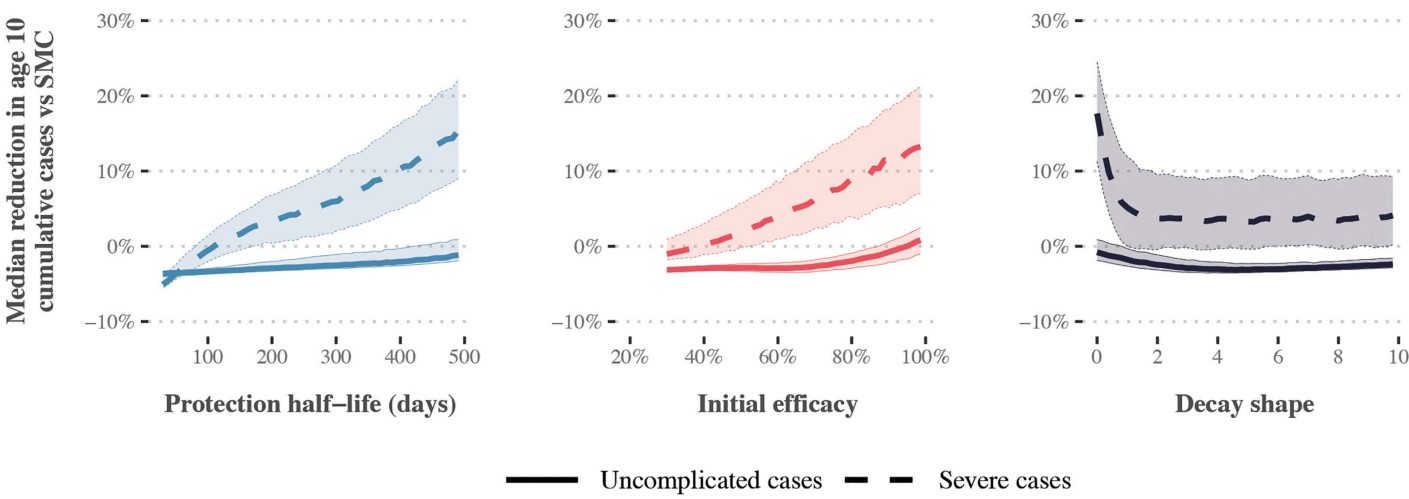

——— Uncomplicated cases – – Severe cases

**Fig 5. Gaussian Process regression emulator predictions for the relationship between blood stage therapeutic properties and expected reductions in cumulative uncomplicated and severe cases relative to SMC alone.** Results show the imperfect deployment scenario, where children aged three to 59 months received three SMC cycles. Transmission is high (32% $Pf$PR$_{2-10}$), 75% of malaria cases occur within six months of the year, and the probability of seeking first-line treatment for clinical malaria over 14 days is low (10%). Each panel shows the median reduction in cumulative uncomplicated (solid lines) or severe cases (dashed lines) achieved by combining seasonal deployment of a blood stage therapeutic with SMC, relative to cumulative cases when SMC is deployed alone. Median reductions are calculated by dividing the parameter range for the therapeutic property shown on the x-axis (protection half-life, initial efficacy, decay shape) into 51 segments, and calculating the median outcome for each given segment across all other parameter values. Shaded regions represent the 25th and 75th percentiles of the corresponding reductions. Emulator predictions are shown separately for cumulative case outcomes at five (panels A) and ten years old (panels B). SMC = seasonal malaria chemoprevention.

**Combining SMC with seasonal deployment of a multi-stage therapeutic**

Finally, results suggest that multi-stage therapeutics with both pre-liver and blood stage activity may be preferred for interventions that aim to reduce the burden of severe malaria in children. When three SMC cycles were deployed with a multi-stage therapeutic with both pre-liver and blood stage activity, we found that pre-liver stage activity was the primary driver of impact on cumulative uncomplicated cases. More than 88% of variation in uncomplicated cases measured by five years old was attributed to the combined impact of the pre-liver stage component's initial efficacy and protection half-life (Fig 6, panel A), and more than 75% of variation in cumulative uncomplicated cases measured by ten years old. However, blood stage activity was crucial for impact on cumulative severe cases by the time children were ten years old; up to 89% of variation in the reduction in cumulative severe cases at ten years could be attributed to the blood stage component's initial efficacy and protection half-life (Fig 6, panel A).

Across different transmission intensities, we found that the main trends remained consistent: pre-liver stage properties drove most of the effect on uncomplicated cases, and blood stage properties were important for impact on severe cases, particularly by ten years of age. However, there were some differences at moderate transmission levels. For example, at 33% $PfPR_{2-10}$, 19% of the variation in reductions in uncomplicated cases by ten years old was explained by the decay shape of the pre-liver stage component, compared with 10% or less at higher transmission levels (Fig 6, panel A). Similarly, blood stage properties contributed less to reductions in cumulative severe cases by ten years old when transmission was 33% $PfPR_{2-10}$ or lower, than they did in very high transmission settings. These patterns were consistent across different assumptions about access to care and seasonality (Fig N in S1 Text).

Importantly, model results indicated that deploying a therapeutic with combined pre-liver stage and blood stage activity could lead to greater impact than either therapeutic component alone. When children received three SMC cycles, the expected gain in median outcome between therapeutics with pre-liver stage activity alone and both pre-liver stage and blood stage activity was up to 22.5% across levels of transmission intensity for reductions by five years old, and up to 9.4% for reductions by ten years old (Fig 6, panel B). This trend occurred across SMC deployment scenarios (Fig O, P, Q in S1 Text). This result suggests that multi-stage therapeutics layered with SMC may offer substantial additional benefit for malaria outcomes in comparison to a single-stage therapeutic.

Interestingly, the added benefit of combining pre-liver and blood stage activity remained stable, or declined slightly, as transmission increased when considering outcomes at five years old (Fig 6, panel B). In contrast, we saw the opposite pattern for outcomes at ten years old: the benefit of including blood stage activity, either alone or as part of a multi-stage therapeutic, was greater in very high transmission settings than in moderate ones.

## Discussion

We used mathematical modelling to estimate the potential benefits of deploying SMC to children under five years in combination with a range of novel prevention therapeutics with both single- and multi-stage targets. We compared the benefits of the combined interventions to either SMC with SP-AQ or to the new therapeutic alone, capturing cumulative patterns in malaria uncomplicated and severe cases by five years of age, during intervention deployment, and at ten years of age, five years after deployment had stopped. To identify the therapeutic profiles that would result in added benefit from combining with SMC over SMC deployed alone, our results showed that developers of pre-liver stage therapeutics will need to prioritise candidates with a long duration (>230 days) and long tail of protection (exponential or biphasic decay). Conversely, we found that combining SMC with a new therapeutic always had added benefit compared with the therapeutic deployed alone. We also demonstrated the benefit of a therapeutic with both pre-liver stage and blood stage activity for impact on severe disease outcomes, highlighting the need for developers to prioritise multi-stage malaria therapeutics in particular.

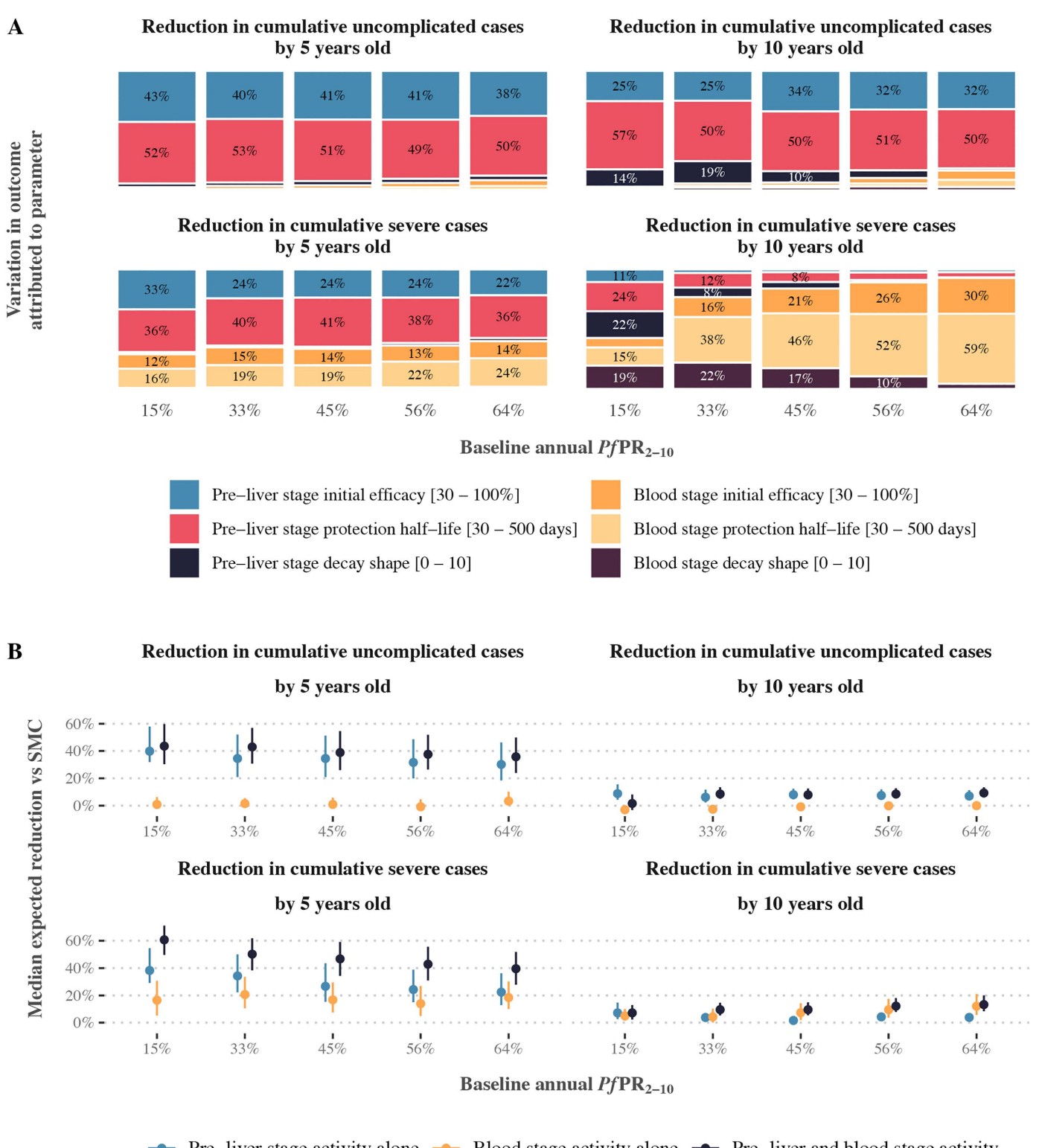

**Fig 6. Impact of combining a therapeutic with both pre-liver stage and blood stage activity with three yearly cycles of SMC on cumulative uncomplicated and severe cases, compared with SMC deployed alone.** Panel A shows the drivers of impact on all cumulative case outcomes for a therapeutic with both pre-liver stage and blood stage activity deployed together with three yearly cycles of SMC, compared with SMC deployed alone.

Bars show the total Sobol effect indices for intervention model parameters. Total Sobol effect indices can be interpreted as the proportion of variation in the cumulative case outcome that can be attributed to a small change in each model parameter and its interactions with other parameters. Indices are shown across levels of transmission intensity (x-axis) for a scenario where three yearly cycles of SMC are deployed (imperfect coverage scenario) in a setting where 75% of malaria cases occur within six months of the year and the probability of seeking first-line treatment for clinical malaria over 14 days is low (10%). Panel B shows differences in the median impact on cumulative case outcomes for therapeutics with pre-liver stage activity alone, blood stage activity alone, and multi-stage pre-liver stage and blood stage activity. Results are shown for the same scenario as in panel A. Points in each panel show the median expected reduction in cumulative cases achieved by combining seasonal deployment of a pre-liver stage, blood stage, or multi-stage therapeutic with SMC, relative to cumulative cases when SMC is deployed alone. Median reductions are evaluated across all possible combinations of therapeutic initial efficacy, protection half-life, and decay shape. Bars indicate the 25th and 75th percentiles of the corresponding reductions. SMC = seasonal malaria chemoprevention, $PfPR_{2-10}$ = Annual *P. falciparum* parasite prevalence in children aged between 2 and 10 years old.

Our results support continued investment in research and development for pre-liver stage malaria therapeutics that meet minimum duration and efficacy criteria. In particular, we found that when a pre-liver stage therapeutic was combined with three cycles of SMC in a high transmission setting, a protection half-life of more than 230 days, an initial efficacy of more than 50%, and a long tail of protection were required to maintain reductions in severe cases by ten years of age. This suggests a likely role for improved malaria vaccines in combination with SMC, rather than a pre-liver stage therapeutic with rapid decay. This work did not specifically attempt to model outcomes for either of the two approved malaria vaccines, RTS,S and R21 and we did not incorporate waning boosting efficacy into our therapeutic models. However, if the goal of deploying SMC in combination with a novel therapeutic is to sustain reductions in severe malaria throughout childhood, our results suggest that these therapeutics will require either extended pre-liver stage duration or multi-stage activity.

Importantly, our findings highlight the public health value of a therapeutic with combined pre-liver and blood stage activity for malaria prevention in children. Model results showed that combining a multi-stage therapeutic with SMC led to greater reductions in cumulative severe cases throughout childhood than single-stage therapeutics. This population-level effect was not the result of any assumed individual-level synergy between intervention components but rather emerged from OpenMalaria's mechanistic structure: each component acts independently on a different stage of the parasite life cycle. Specifically, blood stage activity may treat infections that escape pre-liver stage protection, preventing parasite densities from reaching thresholds associated with severe disease. This aligns with recent findings from the phase two trial of the blood stage vaccine candidate RH5.1/Matrix-M, which reported greater efficacy against high-parasite density infections [36]. Deploying a preventive intervention with blood stage activity may also modify the development of blood stage immunity. Currently, the only approved malaria therapeutics with both liver and blood stage activity are oral antimalarial drugs such as SP-AQ. In the near future, this gap could be filled with the development of long-acting injectable formulation of an antimalarial drug that is active against multiple stages of the parasite life cycle [14–17]. In the longer term, we could expect substantial benefit from layering SMC with a combination vaccine such as RTS,S or R21 combined with an RH5-based vaccine candidate (ClinicalTrials.gov ID NCT05357560) [19], or monoclonal antibody.

Our results also suggest that combining a novel malaria therapeutic with SMC is likely to lead to the greatest public health benefits in settings where there are gaps in SMC's effectiveness. A new vaccine, monoclonal antibody, or long-acting injectable can provide an additional layer of protection for children with poor adherence to SP-AQ, or who are absent for one or more SMC cycles. A new therapeutic can also supplement SMC's protection for the period between cycles when SP-AQ's protection has waned or when the number of SMC cycles does not fully cover the malaria transmission season [37]. In addition, the increased benefit of a blood stage or combination therapeutic for benefit on severe disease outcomes at ten years old may reflect the role of blood stage activity in preventing high-density infections once an SMC program has ended and children are again exposed to frequent infections in high transmission environments.

Although we did not explicitly model resistance to SP-AQ, our results suggest that a novel therapeutic would have its strongest benefits in regions where SP's duration of protection is shortened by resistance. In such settings, children experience longer periods of vulnerability between SMC rounds, increasing both the absolute and relative benefit of an

additional intervention. Reduced SP efficacy would lower the impact of SMC alone, meaning that comparable reductions in malaria burden could be achieved with a novel therapeutic that has a shorter protection half-life or lower efficacy than would be required in fully SP-sensitive settings. As the clinical pipeline of next-generation vaccines and long-acting injectables matures, explicitly evaluating these interactions with varying levels of resistance to SP will be an important area for future modelling.

On the other hand, our results indicate that SMC may not need the support of a new therapeutic when coverage and effectiveness is already high. A longer pre-liver stage protection half-life was needed for a novel therapeutic to contribute additional reductions in malaria burden when children received five cycles of SMC. This result is consistent with previous modelling studies; modelling has indicated that synergisms between a pre-liver stage vaccine and mass drug administration are lost when coverage is high [38], and we have previously demonstrated that increasing SMC's coverage through, for example, the deployment of a new oral chemoprevention drug can lead to substantial increases in SMC's effectiveness [31]. These findings suggest that as the implementation of SMC improves, or should next-generation therapeutics for SMC become available, the added value of layering a second therapeutic may become smaller.

These trade-offs between SMC's coverage and the benefits of a partner therapeutic imply that the cost-effectiveness of a new therapeutic will depend on the public health effectiveness of existing interventions. The number of children missed by SMC varies across programs. For example, the ACCESS-SMC project found that SMC coverage varied between countries; as few as 24% of children in Chad and as many as 86% of children in Burkina Faso received all four cycles of SP-AQ [5]. We did not model cost-effectiveness outcomes in this work, due to the uncertainties inherent in costs for therapeutics that are still being developed. However, our results indicate that adding seasonal deployment of a pre-liver stage vaccine to SMC may not be cost-effective when SMC's coverage has been optimised and the efficacy of SP-AQ remains high. Our results also highlight the important conceptual difference between evaluating an intervention with respect to its absolute number of malaria cases averted and relative to the standard of care; any assessment of intervention cost-effectiveness should be coupled with a strong understanding of the absolute malaria burden in a setting.

All modelling studies are limited by their assumptions. In this study, we measured impact on the cumulative incidence of uncomplicated and severe cases throughout childhood. These outcomes do not capture the potential benefits of malaria chemoprevention for non-malarial health outcomes, and depend strongly on our transmission model's assumptions about the relationships between infection density, malaria immunity acquisition, and the development of clinical disease [23,24]. Future studies of intervention combinations with data from clinical trials and with alternative health outcomes will be important to confirm the findings of our work, and validate or support changes to model assumptions on anti-parasite and anti-disease immunity.

In addition, we made assumptions within our model structure to represent the dynamics of our malaria preventive therapeutics. The model structures used have been previously validated to data for pre-liver stage vaccines and blood stage antimalarial drugs, having been used to estimate the expected public health impact or RTS,S [29] and SMC [26,31]. However, to date, no blood stage vaccines, monoclonal antibodies, or long-acting injectable drugs have been approved by the World Health Organization for use in malaria prevention. Hence, there is limited data available to validate a model structure for the effects of a blood stage therapeutic; new modelling evidence will be needed as blood stage and multi-stage therapeutics advance through clinical trials. We also did not model transmission-blocking therapeutics, as SMC only targets young children and our outcomes focused on malaria burden in children under ten. As a result, the broader community-level benefits of a therapeutic with transmission-blocking activity would not be fully captured. Future modelling studies should assess the potential impact of multi-stage therapeutics with transmission-blocking activity, particularly in interventions targeting all age groups.

Finally, our estimates of benefit were based on outcomes simulated from a limited number of deployment modalities and endemic settings. In particular, we focused on evaluating the impact of combining a novel therapeutic with SMC in high transmission settings, where new tools are most needed to reduce the burden of malaria.

Future modelling studies could explore combinations in perennial settings and for other use-cases and target age-groups. We also did not investigate how sex, gender, or ethnicity would influence our study outcomes, due to the limited data available for validating estimates of intervention effectiveness across these different groups. As a result of these limitations, our work does not offer forecasts of likely therapeutic impact when combined with a chemoprevention program. We instead aim to provide evidence to support investment decisions and inform candidate selection.

In conclusion, this study has provided quantitative evidence to articulate the benefits of new malaria prevention therapeutics during childhood. Our evidence provides funders and policymakers with new understandings of how interventions can be combined to provide added benefits over the standard of care before clinical trials. Results demonstrate the importance of prioritising co-deployment of a novel therapeutic in settings with limited coverage or effectiveness of SMC and deprioritising when the effectiveness of SMC has already been optimised. For the first time, our results demonstrate the potential for a multi-stage malaria therapeutic to improve a program's ability to maintain benefits for children in the years after the deployment of an intervention has stopped.

## Supporting information

**S1 Text. Methods and additional results.**
(PDF)

## Acknowledgments

We wish to thank Dr Jean-Luc Bodmer (Gates Foundation) for his contribution to this project. We would also like to thank Dr Thiery Masserey for his project management support and acknowledge support and advice from all members of the Disease Modelling research unit of the Swiss Tropical and Public Health Institute. Calculations were performed at sciCORE (http://scicore.unibas.ch/) scientific computing centre at the University of Basel.

## Author contributions

**Conceptualization:** Lydia Braunack-Mayer, Melissa A Penny.

**Data curation:** Lydia Braunack-Mayer.

**Formal analysis:** Lydia Braunack-Mayer.

**Funding acquisition:** Melissa A Penny.

**Investigation:** Lydia Braunack-Mayer.

**Methodology:** Lydia Braunack-Mayer, Melissa A Penny.

**Project administration:** Sherrie L Kelly, Melissa A Penny.

**Resources:** Melissa A Penny.

**Software:** Lydia Braunack-Mayer, Josephine Malinga, Narimane Nekkab, Melissa A Penny.

**Supervision:** Jörg J Möhrle, Melissa A Penny.

**Validation:** Lydia Braunack-Mayer, Josephine Malinga, Narimane Nekkab, Melissa A Penny.

**Visualization:** Lydia Braunack-Mayer.

**Writing – original draft:** Lydia Braunack-Mayer.

**Writing – review & editing:** Lydia Braunack-Mayer, Josephine Malinga, Narimane Nekkab, Sherrie L Kelly, Jörg J Möhrle, Melissa A Penny.

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
