## [Decision Letter · Decision Letter 0]

10 Nov 2025

PCOMPBIOL-D-25-01802

Combining seasonal malaria chemoprevention with novel therapeutics for malaria prevention: a mathematical modelling study

PLOS Computational Biology

Dear Dr. Penny,

Thank you for submitting your manuscript to PLOS Computational Biology. After careful consideration, we feel that it has merit but does not fully meet PLOS Computational Biology's publication criteria as it currently stands. Therefore, we invite you to submit a revised version of the manuscript that addresses the points raised during the review process.

We look forward to receiving your revised manuscript.

Kind regards,

Jennifer A. Flegg

Section Editor

PLOS Computational Biology

Jennifer Flegg

Section Editor

PLOS Computational Biology

**Journal Requirements:**

At this stage, the following Authors/Authors require contributions: Lydia Braunack-Mayer, Josephine Malinga, Narimane Nekkab, Sherrie L Kelly, Jörg J Möhrle, and Melissa A Penny. Please ensure that the full contributions of each author are acknowledged in the "Add/Edit/Remove Authors" section of our submission form.

**Reviewers' comments:**

Reviewer's Responses to Questions

**Comments to the Authors:**

Reviewer #1: The authors should calculate the R0 for the different intervention scenarios as well

Reviewer #2: The reviewer thanks the authors for an excellent manuscript, presenting relevant findings on profiles for new therapeutics when combined with SMC based on a mathematical modeling study. There are only a few questions and minor comments for the authors to address, listed below.

Article summary

The manuscript describes results from a mathematical modeling study that evaluated the health benefits of SMC in combination with new therapeutics (vaccines, monoclonal antibodies, and long-acting injectables), targeting multiple stages of the parasite life cycle, across hypothetical transmission settings during and 5 years after interventions, targeted to children under the age of five years, stopped. The author’s results highlight important parameters such as required length of protection for varying SMC schedules, and show that multi-stage therapeutics can increase the effectiveness of SMC.

Notably, results describe required profiles of such therapeutics to be most effective in combination with SMC, and that if SMC reaches high coverage and adherence to all cycles, the added benefit will be marginal. This is relevant for considerations on assessing cost-effectiveness of a new therapeutic, which will depend on the public health effectiveness of existing interventions (such as SMC).

Overall the manuscript is very well written, clearly describing the methods and results. The figures effectively show the results from the exhaustive parameter explorations, providing many important insights on the most relevant parameters, and parameter thresholds. The figure captions are very clear and helpfully include notes for correct interpretation of metrics a reader might be less familiar with (i.e. of the Sobol effect).

The authors also address important limitations in the discussion, such as SP-AQ drug resistance , waning immunity of the vaccine booster, or transmission blocking therapeutics. Of those, it would have been most interesting to see scenarios under varying SP-AQ resistance profiles. The authors provide a hypothesis in the discussion section (L414-416), but this could also have been backed up by simulations, or elaborated on further.

Major comments/questions

- How does transmission intensity influence the results? Figure 4.4, includes variation by transmission intensity, but authors do not pick up on this in the discussion. It seems to be worth mentioning at least that the trends remain mostly constant across transmission settings/ does that hold if the treatment coverage were higher or seasonality longer? Interestingly, for cumulative severe cases 10 years old, blood stage therapeutics effects compared to SMC increases by transmission intensity , but not in 5 year olds.

- Relatedly, how do seasonality and health system scenarios influence the results? I.e. Figure 4.2 would be interesting to see with additional bars (i.e. not filled dots) for case management or seasonality. Figure 2.7 does show both levels for seasonality and also for health system scenarios, and would benefit from additional discussion. It is plausible that with imperfect deployment, the added benefit of therapeutics is larger if seasonality is longer, but this shows only if access to care is high, not low. It would be useful if the authors could elaborate on this in the discussion.

- The supplementary figure 2.8 shows the predicted relationships between pre-liver stage protection half-life, initial efficacy and reductions in cumulative severe cases by ten years old, compared across SMC deployment scenarios for a product with a rapid decay in protection. The figure suggests a slight disadvantage (-5) compared to SMC alone? This is not commented on in the discussion. As it is clear from the discussion, that long acting therapeutics are to be prioritized for combination with SMC, the importance of avoiding rapid decay, even if half life is long, could benefit from further elaboration in the discussion (i.e. L420) and reference in the main text. As it is now, the decay shape is not discussed in the discussion section.

- Looking forward, authors cite previous work they have published on next generation SMC scenarios, could the authors comment how results may change when considering next generation SMCs, or mention this as an area of further research and consideration given that next generation SMCs are also in development.

Minor comments

- In the methods high transmission intensity is described as 64%, and not 32% which is also labelled high. Perhaps in the methods 64% could be described as very high, for distinction of those intensity levels? Relatedly, in Fig 4.4, prevalence levels include 33, instead of 32 mentioned in the main text, and is only starting at 15%, not 9% which is used in the methods.

- L32-33: the authors write that “highlighting the need to prioritise the clinical development of these therapeutics for combination with malaria chemoprevention” , or should it rather be that “clinical development should prioritise agents with demonstrated suitability for combination with malaria chemoprevention”. Whether to prioritise clinical development, or whether to prioritize specific therapeutics during clinical development is a slightly different nuance.

- L54: could consider updating to estimates from latest World Malaria Report

- L150-153: To what do 1) and 9) refer to?

- L210: how many stochastic replicates were run? - not mentioned in the main text, but in the supplement.

- L220: the text mentioned 4 seasonal profiles which appears to be a typo and it should be 2

- L141: to what do the explicit models of therapeutic activity refer to ? to the 500 parameters samples?

- FIg 2B, are the illustrative cumulative cases based on a simulation with transmission intensity corresponding to PfPR2-10 32% ?

- The random allocation does seem to be missing for Predicted cumulative case curves by age in the Supplement.

Reviewer #3: None

**Have the authors made all data and (if applicable) computational code underlying the findings in their manuscript fully available?**

Reviewer #1: Yes

Reviewer #2: Yes

Reviewer #3: Yes

PLOS authors have the option to publish the peer review history of their article (what does this mean? ). If published, this will include your full peer review and any attached files.

**Do you want your identity to be public for this peer review?** For information about this choice, including consent withdrawal, please see our Privacy Policy .

Reviewer #1: **Yes:** Dr. Gbenga Aduagbemi Adegbite

Reviewer #2: No

Reviewer #3: No

**Figure resubmission:**
---

## [Decision Letter · Decision Letter 1]

14 Feb 2026

Dear Prof Penny,

We are pleased to inform you that your manuscript 'Combining seasonal malaria chemoprevention with novel therapeutics for malaria prevention: a mathematical modelling study' has been provisionally accepted for publication in PLOS Computational Biology.

Best regards,

Jennifer A. Flegg

Section Editor

PLOS Computational Biology

Jennifer Flegg

Section Editor

PLOS Computational Biology

Reviewer's Responses to Questions

**Comments to the Authors: 
Please note here if the review is uploaded as an attachment.**

Reviewer #1: The authors have done justice to the subject matter but be it known to the authors that there are individual based models built in platforms like NetLogo where R0 can be calculated. Although based on the author's ability in explaining the transmission intensity properly, which affects the R0, I believe this question has been answered indirectly.

Reviewer #2: The authors have done an excellent job with addressing the reviewers comment. No further comments.

Reviewer #3: The authors have satisfactorily addressed the comments I made.

**Have the authors made all data and (if applicable) computational code underlying the findings in their manuscript fully available?**

Reviewer #1: Yes

Reviewer #2: Yes

Reviewer #3: None

PLOS authors have the option to publish the peer review history of their article (what does this mean? ). If published, this will include your full peer review and any attached files.

**Do you want your identity to be public for this peer review?** For information about this choice, including consent withdrawal, please see our Privacy Policy .

Reviewer #1: **Yes:** Dr Gbenga Adegbite

Reviewer #2: No

Reviewer #3: No

---

## [Editor Report · Acceptance letter]

PCOMPBIOL-D-25-01802R1

Combining seasonal malaria chemoprevention with novel therapeutics for malaria prevention: a mathematical modelling study

Dear Dr Penny,

I am pleased to inform you that your manuscript has been formally accepted for publication in PLOS Computational Biology. Your manuscript is now with our production department and you will be notified of the publication date in due course.

With kind regards,

Judit Kozma
